# Emotional attachment and resistance to change in the use of technology: A study among Chinese university music teachers

Juanjuan Wang[1]*, Fang Huang[2], Timothy Teo[3]

**1** Conservatory of Music, Qingdao University, Qingdao, Shandong, China, **2** School of Education, Shanghai International Studies University, Shanghai, China, **3** Faculty of Education, the Chinese University of Hong Kong, China

* wangjuanjuanqd@126.com

## Abstract

While technology adoption has received widespread attention in teaching, little research has been conducted on the impact of emotional attachment and resistance to change on technology adoption by teachers, especially in the field of music. This study adopted an extended technology acceptance model to examine music teachers' technology adoption in the context of Chinese university teaching. These results suggest that music teachers in the Chinese universities have positive attitudes toward the use of technology in music education. Their technology using intention was based significantly on the perceived usefulness, their own attitudes toward technology, emotional attachment, and resistance to change. These variables explain 89% of the variance in music teachers' technology adoption intention. Additionally, music teachers' emotional attachment related considerably to the perceived usefulness of technology and resistance to change which proved to be a significant precursor to perceived ease of use. Pedagogical suggestions are provided to improve design thinking and technology adoption for music teachers.

## 1. Introduction

Innovation in educational technology has transformed pedagogy and changed teaching beliefs and educational stakeholders' expectations [1,2]. Ample evidence is found in the literature of the numerous benefits that technology brings to teaching and learning in various subjects, such as improving language learning interests and motivation [3,4], and enhancing teaching effectiveness in science and medical education [5]. Therefore, efforts have been made by administrators and education stakeholders in different countries and regions to promote technology adoption in teaching. However, teachers' adoption of technology has not reached a satisfactory level [6,7], specifically, technology use has demonstrated a high access but low use paradox [8]. To understand the reasons, researchers have posited diverse theories

**Data availability statement:** All relevant data are within the manuscript and its Supporting Information files. Also, the data of the study is available from a data repository at https://osf.io/ua9kx/?view_only=

**Funding:** Startup research fund from Shanghai International Studies University (Grant No. 41004918).

**Competing interests:** The authors have declared that no competing interests exist.

and concepts. These include the well-established technology acceptance model (TAM) [9], the unified theory of acceptance and usage of technology (UTAUT) [10], the first-order versus second-order barriers [11] as well as the third-order barriers [12]. However, research findings vary to a large extent, and scholars have suggested that technology adoption studies need to consider contexts, contents, and culture [13,14], given that technology users in different disciplines and cultural backgrounds think and behave differently [8,15,16].

The literature review suggested that technology acceptance studies were quite saturated in science and very common in language teaching contexts. For example, in addition to the key variables in TAM, science teachers in Abu Dhabi considered self-efficacy and expected benefits of using technology [17]; while Chinese English teachers' technology adoption was influenced by constructive teaching beliefs, subjective norms, facilitating conditions [8], policy [7], and cultural values [14]. Comparatively, although the number of studies focusing on music teachers' technology use has increased since the pandemic [18,19], there are few studies examining music teachers in China [19] and therefore deserve further research. Music education has a long tradition of hands-on teaching [20] and teachers may think differently when using technology. Moreover, emotions impact how people think and behave [21,22] and music teachers are likely to be more emotional than teachers in other fields since they are trained to elicit feelings through their performance [23]. Therefore, variables related to emotions need to be considered when examining music teachers' technology use [24–26].

The success of integrating technology into music education lies in the underlying assumption that teachers are willing to adopt technology and are competent users. However, few studies have unpacked Chinese music teachers' diverse attitudes toward technology and their preferences for integrating technology into music teaching. Among these few studies, Zhang et al [19] noted that technological competence significantly impacted music teachers' beliefs (performance expectancy, effort expectancy, hedonic motivation, and social influence), and these further influenced their use of technology. Mei and Yang [27] suggested that Chinese pre-service music teachers generally understand the potential of using technology in music teaching, but they doubted the efficacy of using technology, and therefore showed a weak inclination to use it in future teaching. Although existing studies have provided an understanding of music teachers' attitudes towards technology usage, they have not considered music teachers' emotional attachment and their resistance to change, two constructs yet to be fully examined in the technology acceptance literature.

How they perceive technology use in music teaching and whether they hold positive emotional attitudes toward digital technology in enhancing learning compared with the traditional master-apprentice teaching model deserve investigation. Without understanding the above issues, it is likely that we will miss salient information on how music teachers in the digital age could harness technology to sustain effective music education, especially under unprecedented conditions such as the Covid-19 pandemic.

Therefore, the current study aims to unpack music teachers' perceptions of adopting technology in music teaching, and specifically, the extend to which the proposed research model based on the technology acceptance theory explained music teachers' technology using intentions. Research questions guiding the study are:

(1). To what extent does the research model explain Chinese university music teachers' intentions to integrate music teaching with technology?

(2). Does emotional attachment influence music teachers' intention to use technology?

(3). Does resistance to change influence music teachers' intention to use technology?

## 2. Literature review

### 2.1. Music teaching with technology

Scholars have examined the music teachers' use of technology in education. For example, in the United States, studies suggested music teachers' technology usage is more frequently than their students [28], but they concerned about the available support from technical and pedagogical perspectives, and the feeling of authenticity (e.g., usefulness) when providing one-to-one teaching to students [29]. [30] suggested that pre-service music teachers were proficient in using current and future music technology, but still had low levels of intention to teach a fully technology-based class, due to insufficient instructional time and limited funding and/or access to technology. In Hong Kong, Ho [31] suggested that when technology was carefully planned, well designed and integrated, it can motivate music majors by enhancing the quality of learning. In North Cyprus, Gorgoretti [32] suggested that technology had changed the role of music teachers allowing for better lesson planning and preparation; pre-service music teachers were competent in using various music technologies (e.g., MIDI keyboard, digital piano, speaker), music software (e.g., Finale, Sibelius), and social media (e.g., Facebook, YouTube, Google +), but the accessibility to innovative technology, technical support, and technology integration in real music teaching were still big issues to be resolved.

In the digital age, music education is available to teachers and students through technological tools, social media, and interactive music activities, whether in urban or rural areas [33]. Music teachers choose online resources to organise workshops and live concerts to help students experience and explore different music styles and develop music creativity [34] and numerous technological resources, tools, and software to facilitate the improvement of musical skills. For example, Nichols [35] noted that the music learning software "SmartMusic" offers teacher-developed musical notations and instructions, a set of practice tools (e.g., a metronome), and recording platform. In addition, it allows teachers to implement personalised teaching in large classrooms where students have diverse interests and skills. Recent studies have suggested that music teachers used online videos in classrooms to provide students with relevant and stimulating materials [36], citing the use of social networking sites (e.g., YouTube) to engage learners [37] and illustrate music fundamentals, including scales, chords, the basics of music form [38]. Furthermore, technology has enabled music teaching in and out of school [39], which is of great significance for teachers, especially in challenging circumstances such as the pandemic.

Despite the above-mentioned benefits of using technology in music teaching, teachers still demonstrated uncertainty in terms of using [30,40], due to obstacles including the traditional nature of music teaching model, such as the master-apprentice mode [40], and lack of technical support and training [32]. Even after the Covid-19 pandemic, which is no longer impeding normal teaching and learning, music teachers' technology using intentions and its influential factors requires further research.

### 2.2. Model development based on the TAM

The model in this study was based on the technology acceptance model, TAM. The validity of TAM in explaining users' affinity for technology has been well suggested in different cultures, such as America [10], Europe [41–43] and Asia

[1]. It has also been used in the music teaching context and has validly in explained music teachers' technology adoption [40,44]. Notwithstanding, TAM was still criticized for the limitations of its explanatory function [8,45]. Therefore, we extended the original TAM by adding two rarely researched variables, emotional attachment (EA) and resistance to change (RC) to the research model. The rationale for including these variables is described below.

TAM is known for being one of the most powerful theories for explaining users' inclination towards technology in diverse content and cultural contexts [8,14,46]. In TAM, the two main variables that explain user attitudes and behavioural intentions are perceived usefulness (PU) and perceived ease of use (PEU). PU measures individual's beliefs about the extent to which technology usage can improve productivity at work or in studies. PEU indicates individuals' beliefs about whether and to what extent technologies are effortless. These two main elements influence attitudes toward technology use, ATU, which indicates individuals' likeness or preference of using technology, and their behavioral intention (BI) that shows their willingness to work with technology [9]. In summary, TAM specifies that intention is explained by attitude and perceived usefulness. Attitude is associated with usefulness and perceived ease of use, which explains usefulness.

The above description of the TAM gives rise to the following hypotheses:

H1: Perceived usefulness significantly and positively influences attitudes.
H2: Perceived usefulness significantly and positively influences behavioral intention.
H3: Perceived ease of use significantly and positively influences perceived usefulness.
H4: Perceived ease of use significantly and positively influences attitude.
H5: Attitude significantly and positively influences behavioral intention.

## 2.3. Emotional attachment

Technology-enhanced teaching and learning enables teachers, students, and technologies to develop relationships of attachment through object-to-person interactions. As a cognitive and emotional bond that connects an individual to technology, emotional attachment, EA, indicates the degree to which an individual user feels strong emotional connections with technology when using it [43]. It is usually target-specific and its measurement needs to be contextualized in a specific situation [43]. In addition, this bond is dynamic over time and varies in strength. Although EA, as a key factor examining whether and how individuals' perceptions are related to their behavior, has traditionally been used to analyze person-to-person relationships, it has rarely been used to account for object-to-person interactions. In educational settings, there are few studies involving EA to explain users' technology adoption [47]. Researchers [43] bridged this research gap by contextualizing the study in a Spanish university and incorporating EA to examine educational users' predilection towards technology; however, the participants were pre-service teachers (university students), which limited the results to the narrow representation of university teachers' perceptions. Therefore, this study contributes to existing literature by empirically testing the associations between EA and key TAM variables to measure their influence on university music teachers' adoption of technology in teaching practices. Previous studies have suggested that EA significantly influences PU and BI [43]. In the digital age, music teachers may become emotionally attached to technologies in their daily lives and work, considering they are using technology every day. They may consider technology useful because of their personal fondness for it and form a strong willingness to use it.

H6: Emotional attachment significantly and positively influences perceived usefulness.
H7: Emotional attachment significantly and positively influences behavioral intention.

## 2.4. Resistance to change

In educational context, technology integration entails diverse teaching practices that enable the utilization of technological devices. This demands a change in teachers' existing methodologies and thus may induce stress causing opposition among teachers [48]. Although university music teachers possess rich professional experience in pedagogy and

curriculum development, the use of technology may still be a challenge for those who prefer traditional drill-and-practice teaching methods. RC refers to the perceived difficulty of breaking a routine or existing ways of doing when an individual faces changes [49]. The technology acceptance literature recognizes it as one of the inhibiting factors that negatively affects an individual's acceptance [50]. Although the role of RC in technology acceptance has been examined in some fields, for example, medicine [49] and organizational management sciences [51], insufficient attention has been paid to understanding the role of RC in teaching with technology. Therefore, the extent to which RC influences teachers' technology adoption remains unclear. In the digital age, many teachers formed their beliefs that using technology is a norm in teaching, they were accustomed to using technology that they were familiar with, but may not be unwilling to learn diverse tools or switch to different tools. Given the nature of this construct, it is reasonable to expect RC to influence music teachers' PEU and intention to adopt technology for teaching [43,52].

H8: Resistance to change significantly influences perceived ease of use.
H9: Resistance to change significantly and negatively influences behavioral intention.

## 3. Research design

This research was approved by the IRB of School of Education at the Shanghai International Studies University (SISUGJ202401). A quantitative approach with data collected from a survey questionnaire is adopted to understand Chinese university music teachers' technology adoption. Informed consents were received from participants who are willing to participate in the study. The details of the method, including participants, instruments, procedures, and data analysis, are presented in the following paragraphs.

### 3.1. Participants

The quantitative study comprised 191 university music teachers. Among the participants, the majority (95.3%) were from universities with normal rankings (non-key/ non-top-ranking universities), with female teachers taking up a larger proportion (62.8%). On average, the participants were aged at 35.5 (SD = 9.548) and had 10.8 [SD = 8.69] years of teaching experience. Regarding teaching with technology, they had an average of 5.77 [SD = 6.02] years of experience. According to their responses, tools that music teachers use for online teaching include *DingTalk*, Tecent Meeting, QQ, WeChat, etc. Of the 192 responses, one was excluded from the data analysis because of incomplete and unverifiable information. Table 1 presents the participants' personal details.

### 3.2. procedure

Using the convenience sampling and snowballing sampling, we have collected data from music teachers at 52 Chinese universities, 20 in South China and 22 in Northern China, at the beginning of 2024. Initially, some music teachers were contacted by the authors and subsequently, more teachers were invited to participate in this project. Before we collect data, we have fully explained to the potential participants regarding the aim of the study, the recruitment criteria (in-service music teachers who had technology using experience in music teacher). After the authors informed the participants of the details of the study, an online survey tool (WeChat) was sent to collect their responses. Generally, the participants averagely10 minutes completing the questionnaire.

### 3.3. Questionnaire survey

An online survey was conducted to collect responses from participants. The first section inquired about participants' demographic information (e.g., age, gender, technological tools, etc.). The second part contained additional information on for the research model (Fig 1). In order to fulfil this study's purpose of understanding university music teachers' perspectives about technology in teaching, the survey items were selected and adapted from diverse sources where their reliability and validity were tested.

**Table 1. Participants' demographic information (N = 191).**

| | Categories | Number/Mean | Percentage |
|---|---|---|---|
| University Type | Key university | 9 | 4.70% |
| | Non-key university | 182 | 95.30% |
| | | | |
| Gender | Male | 71 | 37.20% |
| | Female | 120 | 62.80% |
| | | | |
| Professional title | Lecturer | 82 | 42.70% |
| | Associate Professor | 37 | 19.30% |
| | Professor | 9 | 4.70% |
| | Not given | 63 | 33.00% |
| | | | |
| Age | Range: 23~66 | 35.5 (SD = 9.548) | |
| Yeas of teaching | Range: 0~40 | 10.08 (SD = 8.69) | |
| Years of teaching with technology | Range: 0~40 | 5.77 (SD = 6.02) | |

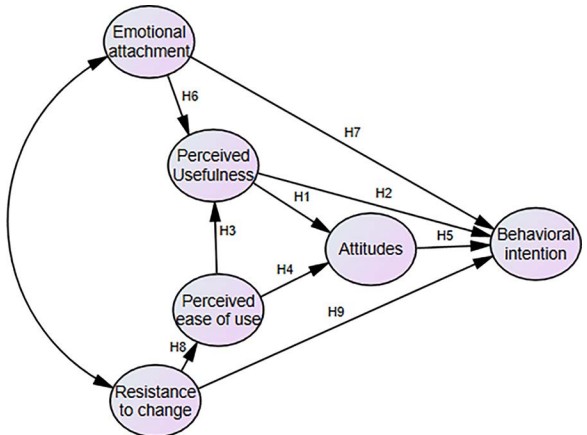

**Fig 1. Illustrates the research model that encapsulates the relationships among all variables included in this study.**

Perceived usefulness measures individual's beliefs about the extent to which technology usage can improve productivity at work or in studies. It encompasses four items, adapted from Davis [9]. The sample item is "I think using technology improves my music teaching performance".

Perceived ease of use indicates individuals' beliefs about whether and to what extent technologies are effortless. It includes four items that were adapted from Davis [9]. The sample item is "I think it is easy for me to use technology in music teaching".

Attitude towards using indicates an individual's fondness of using technology, and it has four indicators, which are also adapted from Davis [9]. The sample item is "Using technology in music teaching is a wise choice".

Behavorial intention measures the degree to which an individual is willing to use technology for given tasks, and it encompasses four items adapted from Davis [9] and the sample item is "I will use technology in my future music teaching".

Emotional attachment indicates the degree to which an individual user feels strong emotional connections with technology when using it [43]. It has five items that were adapted from Sánchez-Prieto et al. [43]; the sample item is "I would feel bad if I cannot use technology".

Resistance to change refers to the perceived difficulty of breaking routines or existing ways when an individual faces changes [49]. It has four indicators adapted from Sánchez-Prieto et al. [43]. A sample item is "I do not expect to change my teaching method".

A 7 -point Likert scale was used in the measurement. To ensure the appropriateness of the survey items, we tested the reliability and validity of the items in the structure equation modeling and specifically, the confirmation factor analysis procedure. Table 2 showed the details of the constructs.Reliability and validity were ensured by the composite reliability (CR) as well as the average variance extracted (AVE). Specifically, the CRs varied from.910 to.968 and AVEs varied from.717 to.882, indicating suitability and construct validity and reliability, according to Fornell and Larcker [53]. Since music teachers in China lack sufficient English proficiency, the items used in this study (originally designed in English) were translated to Chinese. To ensure consistency of meaning, two research assistants who are bilingual in Chinese and English translated the items from English into Chinese, and two other research assistants performed the reverse translation.

**Table 2. Results of the measurement model.**

| Constructs | Items | Standardized factor loading | CR | AVE | Cronbach's alpha |
|---|---|---|---|---|---|
| Perceived usefulness (PU) | PU1 | 0.852 | 0.937 | 0.788 | 0.930 |
| | PU2 | 0.927 | | | |
| | PU3 | 0.884 | | | |
| | PU4 | 0.886 | | | |
| Perceived ease of use (PEU) | PEU1 | 0.876 | 0.91 | 0.717 | 0.909 |
| | PEU2 | 0.875 | | | |
| | PEU3 | 0.829 | | | |
| | PEU4 | 0.804 | | | |
| Attitudes (ATU) | ATU1 | 0.862 | 0.959 | 0.855 | 0.957 |
| | ATU2 | 0.93 | | | |
| | ATU3 | 0.98 | | | |
| | ATU4 | 0.922 | | | |
| Behavioral intention (BI) | BI1 | 0.91 | 0.968 | 0.882 | 0.967 |
| | BI2 | 0.93 | | | |
| | BI3 | 0.955 | | | |
| | BI4 | 0.96 | | | |
| Emotional attachment (EA) | EA1 | 0.925 | 0.938 | 0.753 | 0.939 |
| | EA2 | 0.951 | | | |
| | EA3 | 0.868 | | | |
| | EA4 | 0.747 | | | |
| | EA5 | 0.834 | | | |
| Resistance to change (RC) | RC1 | 0.940 | 0.953 | 0.836 | 0.953 |
| | RC2 | 0.937 | | | |
| | RC3 | 0.901 | | | |
| | RC4 | 0.877 | | | |

### 3.4. Data analysis

This study used the structural equation modelling in the AMOS 22.0 to examine the relationships. As Anderson and Gerbing [54] suggested, a maximum likelihood estimation method with the measurement model followed by the structural model was performed. Specifically, the confirmatory factor analysis (CFA) of the measurement model showed the fit between items and underlying constructs. The hypothesized relationships were examined in the structural model analysis. We follow the well-accepted criteria, such as the standardized factor loadings of items (values greater than.7) [55], the composite reliability and the average variance extracted (acceptable values of.70 and.50, respectively) [53]. More details were presented in the results section.

## 4. Results

### 4.1. Descriptive analysis

Besides participants' demographic information (collected from the survey questionnaire) presented in Table 1, findings of the descriptive analysis showed that the means of the constructs varied from 5.17 to 5.96 (SD from 1.27 to 1.72), suggesting Chinese university music teachers' positive responses to the constructs. Skewness and kurtosis were from −1.35 to −.75 and −.34 to 1.63, respectively and thus, a normal distribution of the data was ensured.

Results of the exploratory factor analysis (EFA) using the principle component analysis conducted in the SPSS 24.0 indicated that the value of Kaiser-Meyer-Olkin (KMO) being.925, indicating a good appropriateness of doing factor analysis. The value of Bartlett's test of sphericity being 6046.267($df = 300$, $p < .001$), and the variance explained by the six extracted factors was 82.204%.

### 4.2. The measurement model

Results of CFA indicated, Mardia's coefficient was 509.651, which was lower than 675[$p$ ($p + 2$)], indicating the multivariate normality [56]. All the item loadings were above.7, indicating good validity of the items [55]. In addition, the composite reliability (CR) and average variance extracted (AVE) met the acceptable values of.70 and.50, respectively [53]. Table 2 shows the standardized factor loadings, CRs, and AVEs.

The model fit indices were the chi-square divided by the degrees of freedom, and a value lower than 3 suggesting a good fit. The Comparative Fit Index (CFI), Tucker–Lewis index (TLI), and goodness-of-fit index (GFI) were used, and values were all greater than.90, suggesting good fits [55]. The root mean square error of approximation (RMSEA) as well as the standardized root mean square residual (SRMR) had values lower than.1, suggesting an acceptable model fit [55]. Based on these criteria, the results confirmed a good fit with the data ($\chi^2/df = 2.879$, CFI = .920, TLI = .907, RMSEA = .99 [.91,.108], and SRMR = .05].

### 4.3. The structural model

The results suggested that the structural model achieved an acceptable fit ($\chi^2/df = 2.852$, CFI = .920, TLI = .908, RMSEA = .099 [.90, 10], and SRMR = .05). All nine hypotheses are supported (see Table 3). To be specific, behavioral intention was significantly associated with perceived usefulness (H2), attitude (H5), emotional attachment (H7) and resistance to change (H9). Perceived usefulness had the biggest influence on music teachers' intentions to use technology, with regression weight being.473. In addition, perceived usefulness was significantly explained by perceived ease of use (H3) and emotional attachment (H6), which further influenced attitude (H1). Resistance to change significantly explained perceived ease of use (H8), which further influenced attitude (H4).

The variance in BI explained by variables of ATU, PU, EA, and RC was 89%, indicating that the research model had a high level of explanatory power for music teachers' technology adoption (RQ1). The variance in ATU explained by PU and

**Table 3. Results of hypotheses.**

| Hypotheses | | Standardized regression | Results |
|---|---|---|---|
| H1 | PU→ATU | 0.792*** | Supported |
| H2 | PU→BI | 0.473*** | Supported |
| H3 | PEU→PU | 0.467*** | Supported |
| H4 | PEU→ATU | 0.124** | Supported |
| H5 | ATU→BI | 0.415*** | Supported |
| H6 | EA→PU | 0.473*** | Supported |
| H7 | EA→BI | 0.187*** | Supported |
| H8 | RC→PEU | 0.250*** | Supported |
| H9 | RC→BI | −0.061** | Supported |

*Note:* * $p < .01$; ** $p < .05$; *** $p < .001$

PEU was 74%, the variance in PU explained by PEU and EA was 46%, and the variance in PEU explained by RC was 6%. Table 3 shows the standardised regressions and results of the hypotheses.

## 5. Discussion

Contextualizing in Chinese higher education, this study examined music teachers' technology adoption and investigated the factors affecting their use of technology in music education. The proposed research model is approved to be valid in explaining Chinese music teachers' intentions to use technology in teaching, with 89% of the BI variance. All the hypotheses in the research model were supported. Chinese university music teachers generally held positive attitudes toward technology use, and their intention was significantly influenced by PU, PEU, ATU as well as perceived EA and RC.

In line with abundant studies in the technology adoption literature [9,10], Chinese music teachers believed the usefulness and ease of use of technology in teaching music significantly and positively influenced their attitudes (H1, H4), with standardized regression coefficients being 0.792 and 0.124, respectively, which further resulted in technology adoption (H5, regression coefficient being 0.415). This showed that music teachers' attitudes (fondness of technology) play an important mediating role in their technology adoption, reflecting previous studies of in-service teachers in other subject areas such as English [8,57] and Math [46]. In today's digital age, music teachers should build up positive attitudes towards technology integration in music teaching, since digital native students are usually technology-savvy and they enjoy learning with technology [58]. PEU exerted a significant and positive influence on PU (H3, regression coefficient being 0.467), indicating that when teachers believe that using technological tools is not confusing or complicated, they are more likely to form a perception of usefulness [9,10]. Like previous studies [7,46], PU significantly and positively influenced Chinese music teachers' intentions (H2, regression coefficient being 0.473). This indicates that PU played a critical role in music teachers' technology adoption in teaching. This observation is reasonable when tech innovations are perceived as having widespread benefits in facilitating music teaching and learning [59].

As for the extended variables (EA and RC) in the research model, emotional attachment (EA) has been found to significantly influence music teachers' perceived usefulness (H6, regression coefficient being 0.473) and behavioral intention (H7 regression coefficient being 0.187), and the relations were both positive. As a psychological and social construct, EA indicated the extent to which music teachers worry, feel anxious or depressed if they could not resort to technology in their teaching. The role of EA in influencing individuals' behaviour has not been widely examined in the technology acceptance literature [60], especially in the context of music teaching. This finding enhanced people's understanding of the role of teachers' emotion in affecting behavior. In the digital age, it is common for people to consider technology as important

in their lives and use technological tools to perform different tasks, including teaching and learning. Although technology cannot replace humans in many ways, music teachers have reached a consensus regarding its assistance, usefulness, and indispensability in music teaching. This is true especially when teachers realized that using technology was the only effective way to sustain teaching in emergency remote learning [1]. Music teachers may perceive that using technology in teaching has become the norm in recent years.

Resistance to change (RC) indicates individuals' perceived difficulty in breaking with their routine or emotional stress when they face changes [49]. It also captures whether and how emotions influence people's behavior. Echoing previous studies that emphasized the necessity to consider resistance to change [43], this study suggested a positive relationship between resistance to change and perceived ease of use (H8), and a negative relationship between resistance to change and behavioral intention (H9). During the pandemic, almost all the teachers were required to continue teaching with assistance from technology. When conducting this study, we found many have formed a technology-integrated pedagogy. In this situation, the unwillingness to change the existing pedagogy leads to their perceptions of ease of use the technology. As for the negative relationship between resistance to change and behavioral intention, it was understandable because in-service music teachers have got professional teaching experience and they had sufficient pedagogical and music knowledge to develop teaching and improve their teaching practices. In addition, as mentioned earlier, music teaching, especially folk music style, demonstrates a typical mentoring style and it is difficult to change. They may use relatively easy-to-learn technology as assistance in teaching. If music teachers are not open to the idea of innovating teaching methods or changing their teaching habits, fancy technological tools will put pressure on them. These feelings have further negative effects on the acceptance of technology in teaching music [50].

## 5.1. Contribution and implication

This study makes both theoretical and educational contributions to the literature. The results demonstrate the power of TAM in explaining music teachers' technology adoption. By extending TAM, this study examined the influence of two rarely researched variables (EA and RC) on music teachers' technology using intention.

This study also provided practical implication to teaching and learning. Some music teachers still perceive using technology as challenging, they are suggested to improve technological knowledge and skills to design and integrate technology in teaching activities. Hands-on practice and tutoring are effective and deserve respect. However, considering that online teaching has increasingly become a norm in education worldwide as a result of students' preferences for technology, teacher training institutions are advised to design targeted training programs to facilitate pre-service and in-service music teaching accordingly. Apart from increasing learning opportunities and providing content in training programs, trainers should also provide practical examples and explanations to improve teachers' understanding of technology use in music teaching and strengthen their confidence in using technology [11].

## 5.2. Limitations and suggestions for further study

Despite the fact that the number of music teachers in universities is not as large as that of teachers in other fields (e.g., Math and English), the sample size still falls short of obtaining a full picture of music teachers' technology adoption, and findings may not be generalized. Moreover, some demographic factors such as age, gender, technology using experiences and teachers' location may also influence music teachers' technology adoption. In addition to these, technology tools differ in their functions and affordances [4]. Therefore, further studies could increase the sample size, consider the individual factors, social factors, institutional factors, and technology types, to examine if play moderating role in certain relationships, so as to broaden the understanding of technology use in music teaching. Last but not least, this study did not control personal traits when measuring resistance to change, future study may measure it to know about if personal traits influence resistance to change.

## 6. Conclusion

This study used a quantitative approach to unpack university music teachers' use of technological tools to teach music. Based on the technology acceptance model, two rarely examined constructs, EA, and RC, were added as exogenous variables to explain music teachers' intention to use technology. The results indicated that Chinese university music teachers' intentions to use technology were significantly associated with perceived usefulness, attitude, emotional attachment, and resistance to change. The results confirmed the validity of the extended TAM in explaining Chinese university music teachers' intentions to use technology. The results also provided suggestions for policymakers to design training programs to improve the emotional attachment element to technology and remove their resistance to pedagogical change to enhance technology use in music education.

## Supporting information

**S1 Data. Data_Music Teacher.**
(XLSX)

## Acknowledgments

The authors express sincere gratitude to participants of the study.

## Author contributions

**Conceptualization:** Juanjuan Wang, Fang Huang.

**Data curation:** Juanjuan Wang.

**Formal analysis:** Fang Huang.

**Methodology:** Fang Huang.

**Resources:** Juanjuan Wang.

**Supervision:** Juanjuan Wang, Timothy Teo.

**Validation:** Fang Huang.

**Writing – original draft:** Juanjuan Wang, Fang Huang.

**Writing – review & editing:** Juanjuan Wang, Fang Huang, Timothy Teo.

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
