## [Decision Letter · Decision Letter 0]

18 Feb 2025

Dear Dr. Wang,

Thank you for submitting your manuscript to PLOS ONE. After careful consideration, we feel that it has merit but does not fully meet PLOS ONE’s publication criteria as it currently stands. Therefore, we invite you to submit a revised version of the manuscript that addresses the points raised during the review process.

We look forward to receiving your revised manuscript.

Kind regards,

Andrea Schiavio

Academic Editor

PLOS ONE

3. In the online submission form, you indicated that [Data will be provided upon an appropriate request to the corresponding author.].

5. Please ensure that you include a title page within your main document. You should list all authors and all affiliations as per our author instructions and clearly indicate the corresponding author.

6. Please amend either the abstract on the online submission form (via Edit Submission) or the abstract in the manuscript so that they are identical.

Reviewers' comments:

Reviewer's Responses to Questions

**Comments to the Author**

1. Is the manuscript technically sound, and do the data support the conclusions?

Reviewer #1: Partly

Reviewer #2: Yes

2. Has the statistical analysis been performed appropriately and rigorously?

Reviewer #1: No

Reviewer #2: Yes

3. Have the authors made all data underlying the findings in their manuscript fully available?

Reviewer #1: Yes

Reviewer #2: No

4. Is the manuscript presented in an intelligible fashion and written in standard English?

Reviewer #1: Yes

Reviewer #2: Yes

Reviewer #1: Dear Author,

After the revision of the article, it is considered a suitable work for publication, however, these aspects are suggested below for improvement of the article:

Introduction:

- It would be advisable to include at the end of the introduction the general objective, the specific objectives and the study hypotheses with their corresponding citations from the studies on which the authors have based these hypotheses.

Instruments:

- An item example should be collected for the variables used. In addition, it should be explained whether the validity and reliability of the instrument was assessed as a single instrument or whether it was assessed by factors, taking into account that they are items of different instruments.

Data analysis:

- The explanation of data analysis should be expanded and criteria added to specify whether the results obtained are adequate according to the test performed.

Procedure:

- A procedural sub-section should be added, setting out how the questionnaire was administered, when, how long it lasted and ethical considerations of the study.

Results:

- If the instrument is set up as a single instrument, an exploratory factor analysis should be carried out beforehand. If different instruments are validated, it should be clarified that different confirmatory factor analyses were performed and explain all this information in the section of instruments (by test) and in the section of data analysis.

- Table 3 needs to be explained in more detail.

Discussion:

- The iscusión should be extended, hypothesis to hypothesis and whether it is rejected or accepted and its justification.

References:

- Review the appointments.

- Review references to the journal’s standards.

Reviewer #2: The present manuscript provides a study that focuses on underexplored aspects of technology adoption, particularly within music education, making it a valuable contribution to the field. It introduces emotional attachment (EA) and resistance to change (RC) as additional variables enhancing the Technology Acceptance Model (TAM), offering a novel perspective.

The research questions and hypotheses are well-structured and align with the study's theoretical framework. The study employs structural equation modeling (SEM), a robust quantitative method for validating relationships among variables. The sample size of 191 university music teachers appears reasonable for this type of study.

The paper effectively contextualizes its findings within previous research and provides practical recommendations for improving technology adoption among music teachers. The discussion highlights pedagogical and policy implications, which can be useful for stakeholders in music education.

Overall, the manuscript is well-written, with a clear and coherent flow of ideas.

However, some aspects require further clarification:

1.Sampling and Participant Characteristics

The study states that convenience sampling was used to recruit participants from 52 Chinese universities, but more details on the recruitment process would be beneficial.

o Were there any inclusion or exclusion criteria?

o Certain factors that could influence the results, such as teachers’ age, years of teaching experience, and years of teaching with technology, do not seem to have been considered. While these aspects are mentioned as descriptive variables, their potential impact on technology acceptance is not discussed. Although this may fall outside the primary goal of the study, acknowledging these as possible influencing factors would strengthen the analysis.

o Additionally, was regional diversity taken into account (e.g., differences between South and North China)? This factor could also play a role in technology acceptance.

2.Definition and Scope of Technology in Music Teaching

The study discusses various ways in which technology can enhance music teaching—ranging from specialized software to social networks, and remote teaching (tele-teaching). However, it is not clear which specific types of technology the study refers to.

o Different technological tools may serve different teaching purposes, and their levels of acceptance may vary accordingly.

o In particular, for the administered questionnaire, it would be helpful to specify which technologies were considered and what the participants were asked to evaluate.

Providing clarification on these points would further strengthen the study’s contribution and make its findings more interpretable.

**Do you want your identity to be public for this peer review?** For information about this choice, including consent withdrawal, please see our Privacy Policy

Reviewer #1: No

Reviewer #2: No

---

## [Author Response · Author response to Decision Letter 1]

4 Mar 2025

We note that the grant information you provided in the ‘Funding Information’ and ‘Financial Disclosure’ sections do not match. When you resubmit, please ensure that you provide the correct grant numbers for the awards you received for your study in the ‘Funding Information’ section.

Response: We have revised the grant information, thank you for your reminding.

All PLOS journals now require all data underlying the findings described in their manuscript to be freely available to other researchers, either 1. In a public repository, 2. Within the manuscript itself, or 3. Uploaded as supplementary information. The data should be provided as part of the manuscript or its supporting information, or deposited to a public repository.

Response:We have provided data as supporting information in the system.

PLOS requires an ORCID iD for the corresponding author in Editorial Manager on papers submitted after December 6th, 2016.

Response:The ORCID of the corresponding author (0009-0008-8564-5452) is provided in the text we submitted, thank you.

Please ensure that you include a title page within your main document. You should list all authors and all affiliations as per our author instructions and clearly indicate the corresponding author.

Response: We have added a title page including authors’ information in the main document, thank you.

Please amend either the abstract on the online submission form (via Edit Submission) or the abstract in the manuscript so that they are identical.

Response: We amended the abstract to make sure they are identical, thank you.

Please include your full ethics statement in the ‘Methods’ section of your manuscript file. In your statement, please include the full name of the IRB or ethics committee who approved or waived your study, as well as whether or not you obtained informed written or verbal consent. If consent was waived for your study, please include this information in your statement as well.

Response: Thank you for your suggestion. We have added contents to describe the IRB and informed consents in the method section (page 12).

Introduction: It would be advisable to include at the end of the introduction the general objective, the specific objectives and the study hypotheses with their corresponding citations from the studies on which the authors have based these hypotheses.

Response: Thank you, we have added the objectives of the study at the end of the introduction. Hypotheses were rationalized in the literature review section, and this is a quite standardized way of presenting hypotheses, so please allow us to remain this style.

Page 5:...Therefore, the current study aims to unpack music teachers’ perceptions of adopting technology in music teaching, and specifically, the extend to which the proposed research model based on the technology acceptance theory explained music teachers’ technology using intentions....

Instruments: An item example should be collected for the variables used. In addition, it should be explained whether the validity and reliability of the instrument was assessed as a single instrument or whether it was assessed by factors, taking into account that they are items of different instruments.

Response: Thank you, we have added item example and descriptions of the validity and reliability of the factors we have used (Page 14).

Data analysis: The explanation of data analysis should be expanded and criteria added to specify whether the results obtained are adequate according to the test performed.

Procedure: A procedural sub-section should be added, setting out how the questionnaire was administered, when, how long it lasted and ethical considerations of the study.

Response: We elaborated on the data analysis, especially the criteria (please refer to revision in the Page 15). In addition, a sub-section named “procedure” was presented following by a section of “participants”, as you suggested (please refer to Page 13).

Results: If the instrument is set up as a single instrument, an exploratory factor analysis should be carried out beforehand. If different instruments are validated, it should be clarified that different confirmatory factor analyses were performed and explain all this information in the section of instruments (by test) and in the section of data analysis.

- Table 3 needs to be explained in more detail.

Response: Thank you, the instrument was set up as a single instrument, we added an EFA result in the text (page 16) . We also explained more about the table 3 in the text (page 18).

Discussion:

- The discusión should be extended, hypothesis to hypothesis and whether it is rejected or accepted and its justification.

Response: We have extended the discussion section (Page 19-21). Thank you for your suggestion.

References:

- Review references to the journal’s standards.

Response: Thank you, references were reviewed.

Sampling and Participant Characteristics: The study states that convenience sampling was used to recruit participants from 52 Chinese universities, but more details on the recruitment process would be beneficial.

o Were there any inclusion or exclusion criteria?

o Certain factors that could influence the results, such as teachers’ age, years of teaching experience, and years of teaching with technology, do not seem to have been considered. While these aspects are mentioned as descriptive variables, their potential impact on technology acceptance is not discussed. Although this may fall outside the primary goal of the study, acknowledging these as possible influencing factors would strengthen the analysis.

o Additionally, was regional diversity taken into account (e.g., differences between South and North China)? This factor could also play a role in technology acceptance.

Response: Thank you, we have used a sub-section titled “procedure” to report the details,including the recruitment criteria. Please refer the Page 13 for the details.

We do agree with your opinion that some demographic factors and location would also influence thinking and behavior, but considering the main aim of the study and the strict word limit, we decided to focus on the roles of emotional attachment and resistance to change in influencing music teachers’ technology adoption.

We do provide suggestions for further study to consider the roles of these factors, please refer to our revision in the text (Page 22).

Definition and Scope of Technology in Music Teaching: The study discusses various ways in which technology can enhance music teaching—ranging from specialized software to social networks, and remote teaching (tele-teaching). However, it is not clear which specific types of technology the study refers to.

o Different technological tools may serve different teaching purposes, and their levels of acceptance may vary accordingly.

o In particular, for the administered questionnaire, it would be helpful to specify which technologies were considered and what the participants were asked to evaluate.

Providing clarification on these points would further strengthen the study’s contribution and make its findings more interpretable.

Response: Thank you for your suggestion. Online teaching is the context we are looking at.

According to teachers’ responses, the tools they use for online teaching mainly include DingTalk, Tencent Meeting, QQ, and WeChat, etc. These are well-acknowledged technology tools in China. We have added this information in the Method section (page 13).

We appreciate your suggestion of thinking of the difference in tools, and we put them in the suggestions for further study (Page 22).

---

## [Decision Letter · Decision Letter 1]

19 Mar 2025

Dear Dr. Wang,

Thank you for submitting your manuscript to PLOS ONE. After careful consideration, we feel that it has merit but does not fully meet PLOS ONE’s publication criteria as it currently stands. Therefore, we invite you to submit a revised version of the manuscript that addresses the points raised during the review process.

We look forward to receiving your revised manuscript.

Kind regards,

Andrea Schiavio

Academic Editor

PLOS ONE

Journal Requirements:

Reviewers' comments:

Reviewer's Responses to Questions

**Comments to the Author**

Reviewer #2: All comments have been addressed

Reviewer #3: (No Response)

Reviewer #4: (No Response)

2. Is the manuscript technically sound, and do the data support the conclusions?

Reviewer #2: Yes

Reviewer #3: Yes

Reviewer #4: (No Response)

3. Has the statistical analysis been performed appropriately and rigorously?

Reviewer #2: Yes

Reviewer #3: Yes

Reviewer #4: (No Response)

4. Have the authors made all data underlying the findings in their manuscript fully available?

Reviewer #2: Yes

Reviewer #3: No

Reviewer #4: (No Response)

5. Is the manuscript presented in an intelligible fashion and written in standard English?

Reviewer #2: Yes

Reviewer #3: Yes

Reviewer #4: (No Response)

Reviewer #2: I thanks the authors to have addressed my comments and provided the requested clarifications.

I believe that the manuscript is now ready for publication in its current version.

Good luck!

Reviewer #3: This manuscript does indeed present original research, and is technically sound. However, there remain some minor issues to be resolved.

First, a link to the dataset is not currently present, and a dataset was not attached with the reviewer copy. There is no indication that these data are publicly available separately from the manuscript. Either way, the authors need to supply a link to replication data and code. If a dataset was supplied and supposed be attached, disregard.

Second, I would suggest the authors put their RQs in bullet point form, and to rephrase RQ1 so it's clearly referencing the Technology Acceptance Model.

Third, the hypotheses are technically correct but uninformative. If the authors expect one variable to have a positive/negative relationship with another, there is nothing wrong with stating that. It doesn't affect the analyses, but it's immensely helpful to the reader. I'm sure the original theory proposed directional hypotheses. I would also suggest using full variable names instead of abbreviations to add clarity.

Fourth, if Figure 2 is the model being tested, the authors do not need Figure 1.

Fifth, when discussing measurement, it would be easier if the authors presented one variable at a time, paragraph by paragraph (similar to how social psychologists would present it). That way, all the information about a variable (number of items, example of an item, reliability) is in one place.

Sixth, it is more common to use Cronbach's alpha or fit statistics from a Confirmatory Factor Analysis to assess how well a variable fits to the data. I would suggest the authors use either one of those when discussing reliability/data fit instead of Composite Reliability or Average Variance Explained.

Seventh, when discussing results, it is helpful to actually state what the results were beyond whether they were significant. What was their direction? Does the finding support the hypothesis? Yes, it can be seen from Table 3, but it is worth stating to the reader verbally as well.

Eighth, the coefficient testing H8 is missing a digit.

Reviewer #4: (No Response)

**Do you want your identity to be public for this peer review?** For information about this choice, including consent withdrawal, please see our Privacy Policy

Reviewer #2: **Yes: ** Francesca Fracasso

Reviewer #3: No

Reviewer #4: No

---

## [Author Response · Author response to Decision Letter 2]

20 May 2025

Response to reviewer #2:

Thank you for your support. We have added Data availability statement in the text (page 25), thank you. We have accordingly change the presenting style into bulletin form (page 5). We clarified the positive or negative relationships in the hypotheses for those developed from theory. We also use the full names for the variables which are easier for the readers to understand. Please refer to our revision in the text. Thank you, we have removed figure 1 from the text. We took your suggestion and present variables one by one, using different paragraphs (page 14). When doing structural equation modeling, it is a norm to use CR and AVE to report reliability and validity. But for readers’ sake, we also reported values of cronbach alpha (page 17). We have revised the discussion section based on your suggestions. Please refer to our revision in the text (page 19-21). Thank you, we have added the digit in the text (page 19).

Response to reviewer # 4:

Thank you. Following your suggestions, We have indicated positive or negative relationships in the hypotheses except H8, since we could not ensure its direction, but the result suggested its positive direction, and we have provided discussion on it (page 22). We have added their operational definitions in both the literature review section (page 10 & 11) and survey section (page 15), in the text. Thank you for your informative suggestion, we revised this part in the text (page 14).We really appreciate your suggestions. We added descriptions to indicate the time of the data collection (page 14 ). It is worth noticing that technology acceptance study entails willingness to use technology, and its value is to measure the degree to which some factors influence people’s thinking, no matter if they have used technology or not. This can be witnessed in many technology acceptance studies in literature. Therefore, we believe it is still proper and valuable to understand teachers’ technology using intentions, and know more about the degree to which some important psychological factors influence their intentions. In the text, we added rationalization for resistance to change, and clarified that it measures their willingness to innovate existing pedagogy (page 11). We did not control personal traits in this study, but yes, we may measure personal traits in future studies to know about if personal traits influence resistance to change. We put it in the limitations and suggestions for future study section (page 24).Thank you, and we followed your suggestions and renamed method to research design. Please allow us to remain “participants”,”Procedure”... as they are also standardized in educational and psychological studies. We ensure that details of ethical approval has been introduced in the research design section (page 12). We have added clarifications in this section (page 14-15). Please allow us to remain descriptive results in the result section as we believe they belong to it, and many studies do so. Technology acceptance examines (1) if people use or not use technology and (2) the degree to which they are willing to use, by using behavioral intention as the endogenous variable, because it directly leads to actual usage (Davis, 1989).

We also understand there is a big difference in terms of using experience, but since the focus of the study using SEM technical is to understand the relationships, instead of gaining deep understanding of differences by cases, we did not group participants in the data analysis. Future studies could do so to see if using experience moderated some relationships. We put it in the suggestions for future study section (page 24). Resistance to change measures their willingness to change existing ways of doing. As innovative technology always emerges, it is appropriate to be included in this study. Thank you, we do agree that gender, institutional infrastructure and resources may influence teachers’ technology using intentions. We prefer to analyze them in the future studies. Sorry for this, we have revised it. Thank you, we have revised the discussion section as suggested. As many studies using SEM as a technique, we presented results align with existing studies. SEM requires the exogenous variables should be correlated when doing analysis. We did not build up a directional relationship between emotional attachment and resistance to change. But in the data analysis, we have to draw correlations between the two variables since SEM technique indicates variables rotate statistically. As for the definition of resistance to change, we have provided elaborations in the text. Thank you for your wonderful insights. We have benefited a lot.

---

## [Decision Letter · Decision Letter 2]

12 June 2025

Emotional attachment and resistance to change in the use of technology: A study among Chinese university music teachers

PONE-D-24-35928R2

Dear Dr. Wang,

We’re pleased to inform you that your manuscript has been judged scientifically suitable for publication and will be formally accepted for publication once it meets all outstanding technical requirements.

Kind regards,

Andrea Schiavio

Academic Editor

PLOS ONE

Additional Editor Comments (optional):

Reviewers' comments:

Reviewer's Responses to Questions

**Comments to the Author**

Reviewer #2: All comments have been addressed

Reviewer #3: All comments have been addressed

2. Is the manuscript technically sound, and do the data support the conclusions?

Reviewer #2: Yes

Reviewer #3: Yes

3. Has the statistical analysis been performed appropriately and rigorously?

Reviewer #2: Yes

Reviewer #3: Yes

4. Have the authors made all data underlying the findings in their manuscript fully available?

Reviewer #2: Yes

Reviewer #3: Yes

5. Is the manuscript presented in an intelligible fashion and written in standard English?

Reviewer #2: Yes

Reviewer #3: Yes

Reviewer #2: (No Response)

Reviewer #3: All of my queries have been satisfied, and I now recommend that this paper be accepted for publication.

**Do you want your identity to be public for this peer review?** For information about this choice, including consent withdrawal, please see our Privacy Policy

Reviewer #2: No

Reviewer #3: No

---

## [Editor Report · Acceptance letter]

PONE-D-24-35928R2

PLOS ONE

Dear Dr. Wang,

I'm pleased to inform you that your manuscript has been deemed suitable for publication in PLOS ONE. Congratulations! Your manuscript is now being handed over to our production team.

Kind regards,

on behalf of

Dr Andrea Schiavio

Academic Editor

PLOS ONE